# Special Low Protein Foods in the UK: An Examination of Their Macronutrient Composition in Comparison to Regular Foods

**DOI:** 10.3390/nu12061893

**Published:** 2020-06-25

**Authors:** Georgina Wood, Sharon Evans, Kiri Pointon-Bell, Júlio César Rocha, Anita MacDonald

**Affiliations:** 1Faculty of Health, Education & Life Sciences, Birmingham City University, City South Campus, Westbourne Road, Edgbaston, Birmingham B15 3TN, UK; Kiri.Pointon-Bell@bcu.ac.uk; 2Birmingham Women’s and Children’s NHS Foundation Trust, Steelhouse Lane, Birmingham B4 6NH, UK; sharon.morris6@nhs.net (S.E.); Anita.Macdonald@nhs.net (A.M.); 3Nutrition & Metabolism, NOVA Medical School, Faculdade de Ciências Médicas, Universidade Nova de Lisboa, Campo Mártires da Pátria, 130, 1169-056 Lisbon, Portugal; rochajc@nms.unl.pt; 4Center for Health Technology and Services Research (CINTESIS), R. Dr. Plácido da Costa, s/n, 4200-450 Porto, Portugal

**Keywords:** phenylketonuria, special low protein foods, nutritional composition, UK, macronutrients

## Abstract

Special low protein foods (SLPFs) are essential in a low phenylalanine diet for treating phenylketonuria (PKU). With little known about their nutritional composition, all SLPFs on UK prescription were studied (*n* = 146) and compared to equivalent protein-containing foods (*n* = 190). SLPF nutritional analysis was obtained from suppliers/manufacturers. Comparable information about regular protein-containing foods was obtained from online UK supermarkets. Similar foods were grouped together, with mean nutritional values calculated for each subgroup (*n* = 40) and percentage differences determined between SLPFs and regular food subgroups. All SLPF subgroups contained 43–100% less protein than regular foods. Sixty-three percent (*n* = 25/40) of SLPF subgroups contained less total fat with palm oil (25%, *n* = 36/146) and hydrogenated vegetable oil (23%, *n* = 33/146) key fat sources. Sixty-eight percent (*n* = 27/40) of SLPF subgroups contained more carbohydrate, with 72% (*n* = 105/146) containing added sugar. Key SLPF starch sources were maize/corn (72%; *n* = 105/146). Seventy-seven percent (*n* = 113/146) of SLPFs versus 18% (*n* = 34/190) of regular foods contained added fibre, predominantly hydrocolloids. Nine percent of SLPFs contained phenylalanine > 25 mg/100 g and sources of phenylalanine/protein in their ingredient lists. Stricter nutritional composition regulations for SLPFs are required, identifying maximum upper limits for macronutrients and phenylalanine, and fat and carbohydrate sources that are associated with healthy outcomes.

## 1. Introduction

In phenylketonuria (PKU), the only UK treatment option is a rigorous low phenylalanine diet that is essential to prevent neurotoxicity and irreversible brain damage [1]. Most patients with classical PKU tolerate < 10 g natural protein daily [2], with up to 80% of daily protein provided by minimal phenylalanine-containing protein substitutes which are derived from either L-amino acids or glycomacropeptide. Special low protein foods (SLPFs) are an integral part of dietary treatment. They contribute essential energy (up to 50% of intake), variety and bulk, helping to improve or maintain metabolic control and growth [3,4,5]. Given their importance in a low protein diet, their nutritional profile and food labelling should receive the same care and attention as regular foods.

The composition and labelling of SLPFs is regulated by European Commission (EC) legislation on “dietary foods for special medical purposes” [6]. It gives no guidance on the source, amount or even quality of the carbohydrate and fat added to SLPFs [6]. The EC and UK regulations require SLPFs to list the amounts of energy, carbohydrate (including sugars), fat, protein and salt per 100 g [6,7,8,9] but no upper nutrient limits are defined. As a consequence of protein removal, it is expected that lower protein foods will contain higher amounts of carbohydrate and possibly fat [4,10,11], but there is no research describing the nutritional composition of UK SLPFs.

Considering that SLPFs receive minimal regulation, and with limited research into their nutritional profile, it has been suggested that a detailed analysis of each country’s SLPFs be conducted [4,10]. The present study aimed to analyse the nutritional composition of all SLPFs available by the Advisory Committee of Borderline Substances (ACBS) prescription system in the UK.

## 2. Materials and Methods

From January–May 2019, detailed nutritional composition data for all UK SLPFs available on ACBS prescription was collected from manufacturers and suppliers. Data was obtained from company websites or from information sheets provided directly from the companies. Nutritional data was obtained per 100 g/100 mL and per serving for cooked and dried weight of products for: energy, protein, phenylalanine, total carbohydrate, sugars, fibre, total fat, saturated fat and salt. If nutritional data was stated as less than a certain value, e.g., “<0.1” or “<0.5”, 0.001 was deducted from these numbers and values of “0.099” or “0.499” were used. Product ingredients, sources of added fibre, starch, sugar, fat and phenylalanine were obtained. Information was stored on an excel spreadsheet. Products were divided into 10 groups in a similar way to Pena and colleagues [10], and included: bread products (bread, pizza bases), pasta/rice/noodles, flour/mixes, meat/meat replacers, breakfast products (cereals and bars), eggs/egg replacers, milk/milk replacers, snacks (biscuits, cakes, crisps, chocolate, rusks, hazelnut spread and crackers), desserts (rice pudding, flavoured desserts, yogurt, and jelly) and other snacks/meals (soups, potato cakes, cheese sauce and potato pots). These groups were then categorised into 40 subgroups of equivalent product types, e.g., burgers, sausages, cookies/biscuits, cake mixes. The mean and range values for every nutrient across subgroups of similar products were calculated.

The same information (except for sources of phenylalanine) was collected and calculated for at least 2 regular protein-containing comparable foods per subgroup, from major UK supermarkets with nutritional analysis data online (ASDA, Morrisons, Sainsburys, Tesco, Waitrose, Ocado and Marks & Spencer). Phenylalanine content was estimated by calculating that 1 g of protein contained 50 mg phenylalanine [12]. Taste, texture, recipe ingredients and food function were considered when choosing comparator foods. Where possible, only regular products that had nutritional analysis available in the same format as SLPFs were considered, e.g., dried format or after preparation. Percentage differences between SLPFs and regular foods for all mean nutritional values were then determined. Variations of ±0–10% were considered comparable.

## 3. Results

One hundred and fifty one SLPFs were identified on UK ACBS prescription. One SLPF was undergoing reformulation and regular comparators for four SLPFs were not available. Thus, 146 SLPFs were compared with 190 regular products. Appendix A displays all SLPF and regular product subgroups (*n* = 40) and the investigated variables.

### 3.1. Energy

Mean energy content (per 100 g) for all SLPFs (*n* = 146) was 292 kcal (range: 32–583 kcal) and for all regular foods (*n* = 190) was 298 kcal (range: 26–558 kcal). Energy content was comparable for 50% of the subgroups of products (*n* = 20/40). For SLPFs, mean energy values for low protein hazelnut spread, prepared sausage mixes, prepared burger mixes, egg white and egg replacers were 37–66% lower than regular varieties. Low protein dessert pots, hot breakfast cereals, potato pots and fish substitutes contained 36–41% more energy than regular versions.

### 3.2. Protein and Phenylalanine

All SLPF subgroups contained between 43–100% less protein and 60–100% less phenylalanine than regular foods. Table 1 displays the mean and range for phenylalanine content and sources of phenylalanine for all SLPF subgroups. The main sources of phenylalanine found in SLPFs were milk (including milk protein) (32% of SLPFs; *n* = 47/146) and yeast (14% of SLPFs; *n* = 21/146). For 91% of SLPFs (*n* = 133/146), the phenylalanine content was either ≤25 mg per 100 g or no sources of phenylalanine/protein were identified in the product ingredient list (Table 1).

### 3.3. Carbohydrate (Including Sugars)

Overall, the carbohydrate content was higher in 68% (*n* = 27/40) of SLPF subgroups when compared to protein-containing foods, with the greatest differences for meat, fish and egg substitutes (281–9167%).

The percentage of foods containing added sugar is given in Figure 1. Only 35% (*n* = 14/40) of SLPF subgroups contained higher amounts of sugar with 45% (*n* = 18/40) containing less than regular foods. Fish substitute contained 1000% more sugar than regular fish, but the amount of sugar was small (sugar content in fish substitute 1.1 g/100 g). Low protein pizza bases, flour and breakfast cereals contained only 3–22% more total carbohydrate than regular foods, but 81–273% more sugar.

Over 70% (72%; *n* = 105/146) of SLPFs compared with 66% (*n* = 125/190) of regular foods contained an added sugar source (Figure 1), with low protein bread, milk and meat replacements commonly adding sugar where regular foods did not. Key sugar sources in both groups are given in Table 2.

Maize/corn and potato starch were the main types of starch used in SLPFs. Over 70% (*n* = 105/146) of SLPFs contained maize/corn starch whereas 56% (*n* = 82/146) included potato starch. Fifty-four percent (*n* = 79/146) of SLPFs contained both starches. Maize/corn starch was common in low protein pasta, rice and noodles (100%; *n* = 43/43) and snacks (80%; *n* = 20/25). In contrast, the most common starch sources identified in regular foods were wheat flour (*n* = 82/190); wheat semolina (*n* = 30/190) and rice or rice flour (*n* = 27/190). Maize/corn starch and potato starch were only listed in 13% (*n* = 24/190) of regular foods.

### 3.4. Total and Saturated Fat

Sixty three percent (*n* = 25/40) of SLPF subgroups contained less total fat (including egg substitutes, meat replacements, flour/mixes, flavoured desserts (dried powder), dried breakfast cereal, pasta, rice and noodles), whilst 28% (*n* = 11/40) contained 21–94% more total fat (including breads, pizza bases, breakfast bars, fruit bars, chocolate, pasta and sauces, risotto, dessert pots, rusks and liquid milk replacers) than regular foods. In 8% (*n* = 3/40) of the SLPF subgroups, total fat content was comparable to that found in regular foods. Calculation of percentage differences between SLPF egg whites and regular egg whites was not possible, due to SLPF egg whites reporting “nil added” for total fat content.

Thirty-five percent (*n* = 14/40) of SLPF subgroups contained more saturated fat (14–262%) than regular foods, including cakes, breakfast bars, pizza bases, fruit bars, bread and breakfast cereals. Conversely, 50% (*n* = 20/40) of SLPF subgroups contained less saturated fat (<−10%) than regular foods. SLPF pizza mixes, cake mixes, eggs and fish substitutes contained 85–100% less saturated fat.

Palm oil was the most common fat source found in 25% (*n* = 36/146) of SLPFs. Twenty-five (17%) of these SLPFs did not specify if palm oil was hydrogenated or non-hydrogenated but one food contained partially hydrogenated palm oil (<1%), one hydrogenated palm oil (<1%) and nine non-hydrogenated palm oil (6%) (Figure 2). Hydrogenated vegetable oil was another common fat source in SLPFs (23%, *n* = 33/146) (Figure 2). SLPFs with “hydrogenated vegetable oil” or “hydrogenated palm oil” were all produced by the same manufacturer and it was unclear if the sources were partially hydrogenated. The most prevalent fat sources in regular foods were milk (41%, *n* = 78/190) and palm oil (39%, *n* = 75/190), with no products listing hydrogenated oil sources (Figure 2). Palm oil was found in 80% (*n* = 20/25) of SLPF snacks compared with 58% (*n* = 23/40) of regular snacks.

In the SLPF subgroups containing less saturated fat (*n* = 20/40), hydrogenated vegetable oil was present in 35% (*n* = 7/20) (cheese sauce, soups, flavoured desserts, pasta and sauces, xPots and meat replacements).

### 3.5. Fibre

From the nutritional analysis, only 44% (*n* = 64/146) of SLPFs quantified a fibre amount compared with 82% (*n* = 156/190) of regular foods. When fibre content was listed, low protein milk (liquid) and egg substitutes contained more fibre than regular comparator foods which did not contain added fibre. Low protein French toast, chocolate, bread, pizza bases, cake mixes and fruit bars contained more fibre (16–189%) than regular foods. The largest differences were for egg white replacers, burger and fish substitutes (1645–5050%), with SLPFs containing higher amounts.

Some products contained natural fibre sources such as whole-wheat flour or apple flakes but only added fibre sources (e.g., barley/wheat/gluten-free wheat fibre, methylcellulose, pectin, guar gum etc.) were identified from the ingredient lists. Added fibre was found in 77% (*n* = 113/146) of SLPFs but only 18% (*n* = 34/190) of regular foods (Figure 3). The main fibre sources added to SLPFs were methylcellulose, guar gum, hydroxypropyl-methylcellulose, inulin and carob/locust bean gum. These were added to primarily improve texture and quality.

### 3.6. Salt

Over 50% of SLPF subgroups contained 17–100% less salt than regular foods (*n* = 21/40), with low protein rice pudding, chocolate and jelly subgroups all containing 100% less. Salt content was higher in 33% of SLPF subgroups when compared to regular foods with higher amounts in low protein potato pots, xPots, hazelnut spread, crisps, cakes, hot breakfast cereal, fish substitute and pizza mix (100–1050%).

## 4. Discussion

This is the first study to investigate the nutritional composition of all SLPFs available on UK ACBS prescription, compared with regular protein-containing foods, examining macronutrients and their ingredient sources. The overall nutrient quality of SLPFs was variable with no consistent pattern. Some of the nutrients reported on food labelling were incomplete with 56% of foods not itemising fibre content. The energy content of 50% of SLPF subgroups was comparable to regular foods, with only 23% of SLPF subgroups containing a higher amount (>10%) than regular foods.

Sixty three percent of SLPF subgroups contain less total fat and 50% contain less saturated fat (<−10%) when compared to regular foods, including: milk powder, eggs, biscuits/cookies, crisps, crispbread crackers, flavoured desserts, yogurt, cheese sauce, soup, potato cakes, meat and certain flour/mixes subgroups. This appears advantageous. Some studies in PKU, have reported improved or similar biomarkers of cardiovascular disease when compared to healthy controls [13,14,15,16,17]. However, although 50% of SLPF subgroups contained less saturated fat than regular foods, some of the subgroups listed hydrogenated vegetable oil as a fat source and did not specify if this was “partially” or “fully” hydrogenated. Full hydrogenation of vegetable oil produces exclusively saturated fats, whereas partial hydrogenation of vegetable oil leads to a higher amount of trans fatty acids [18,19]. Consumption of trans fatty acids has been linked to the development of several health problems, including metabolic syndrome, coronary heart disease, obesity and diabetes [18,19,20]. Although dietary trans fatty acids may have a similar elevating effect on LDL-cholesterol to that of saturated fatty acids, the former will contribute to HDL-cholesterol reduction [21]. Low HDL-cholesterol has already been reported in PKU patients [14]. Therefore, some SLPFs that may appear “healthier” with a low saturated fat content may actually be higher in trans fats, but this information is not disclosed by the manufacturers. In contrast, 35% of SLPF subgroups contained more saturated fat than regular foods, particularly staple items such as breakfast cereal and breads, which is a concern. Common fat sources were palm oil and hydrogenated vegetable oil, both of which contain saturated fat [18,20,22,23]. The chain length of saturated fat is important, with longer-chain saturated fatty acids being more harmful, whilst short- and medium-chain fatty acids have potential benefits on metabolic risk, weight gain, obesity and gut microbiome [24]. In summary, more precise information on the type of fat added is required for SLPFs.

Over 70% of SLPFs on UK prescription contained added sugar but this percentage was only slightly higher than regular foods. When subgroups were examined more closely, it was apparent that certain SLPFs commonly added sugar when regular foods did not. Specifically, 100% of low-protein breads, pizza bases, flour, meats, crackers, flavoured desserts, yogurt, milks and some pastas contained added sugar. Maize/corn and potato starch were the most frequently used starch sources in SLPFs with most ingredient lists indicating that these starches were present in isolation. Isolated starches are more refined than regular flour and/or raw materials, and foods containing isolated starches may have a higher glycaemic index (GI) than those made from wheat flour [25,26]. In contrast, the addition of fat to a regular carbohydrate food is known to delay gastric emptying and lower GI [27]. The GI of SLPFs available on UK ACBS prescription has not been formally evaluated. This needs to be determined as it is uncertain how the isolated starches, added sugar and increased levels of fat found in some SLPFs impact on GI function.

In PKU, a high carbohydrate intake and the carbohydrate profile of SLPFs may contribute to higher levels of insulin resistance, as a relationship between the quality and amount of carbohydrate in SLPFs and peripheral insulin resistance has been reported [11,28]. An association between the overall glycaemic load and triglyceride glucose index in children with PKU has also been described [11]. In patients with increased abdominal obesity (waist circumference), which is a component of metabolic syndrome, increased triglycerides, lower HDL-cholesterol and increased HOMA-IR (homeostasis model assessment of insulin resistance) is documented [14]. Insulin resistance, a marker of metabolic syndrome, is linked to an increased risk of cardiovascular disease [29].

Gluten and other proteins in regular grains/cereals are important in maintaining structural integrity, texture and quality of regular foods [25]. However, with the majority of SLPFs based on maize/corn/potato starches, it is not surprising that 77% of SLPFs contained added fibre, predominantly in the form of hydrocolloids. Hydrocolloids are additives that improve the quality, formulation and texture of low protein and gluten-free products [25,26,30]. Their contribution as a source of dietary fibre has not been explored, despite the fibre content of hydrocolloids typically varying between 60–90% [31]. Generally, such additives are used in small amounts and are commonly not significant enough to make a fibre claim on a product [31]. However, in patients with PKU where approximately 50% of their energy intake may be from SLPFs [3] containing hydrocolloids, it is probable that these ingredients are significantly contributing to daily fibre intake, although this remains unreported. Therefore, regular consumption of SLPFs may also have an impact on gastrointestinal function and gut microbiome, with previous research reporting that 34% of patients with PKU suffer from digestive problems [2].

Over 30% of SLPF subgroups contained more salt than regular foods, with some containing 100–1050% extra. It is possible that their habitual consumption may contribute to nutritional co-morbidities such as hypertension [32,33,34], vascular stiffness [34,35], overweight/obesity [3,34,36,37,38,39,40] and an atherogenic lipoprotein profile [34].

For 91% of SLPFs, phenylalanine content was ≤25 mg/100 g of the product, or all product ingredients were “exchange-free”, meaning these items can be eaten without measurement [41]. The remaining 9% of SLPFs contained phenylalanine >25 mg/100 g and included ingredients such as milk and potato flakes; and consequently, these foods must be restricted and given in controlled amounts in a low phenylalanine diet [41]. The few SLPFs containing >25 mg/100 g add complexity to a low phenylalanine diet as patients and caregivers may be unsure about their suitability.

Overall, there is limited research into the dietary patterns of patients with PKU, but evidence suggests that SLPFs contribute up to 47% of energy intake [11]. Many contemporary low phenylalanine protein substitutes have a low fat and carbohydrate content, meaning there is an increased reliance on SLPFs to provide these macronutrients [42,43]. With a “treatment for life” policy, it is essential that SLPFs have a nutritional profile that supports long term healthy eating patterns.

There are many recommendations required to improve standards in the nutritional composition and labelling of UK SLPFs. Transparency is necessary by SLPF manufacturers about the nutritional profile of their products. All ingredients should be clearly listed including sources of, at least, starch, sugar, fat and fibre and the amount of fibre added (per 100 g/100 mL) for all SLPFs. Nutritional analysis for both dried and prepared weights should be available. Packaging and website nutritional information should be accurate and consistent. To ensure that all SLPFs can be safely consumed without calculation and measurement, the phenylalanine content should be no more than 25 mg/100 g for all prescribed SLPFs; and no more phenylalanine than 5 mg/100 mL for milk replacements [44]. SLPF macronutrient composition regulations should be strengthened, ensuring similarity to regular protein-containing comparators. Upper limits should be set for carbohydrate and fat content. Fat sources should be predominantly poly- or mono-unsaturated rather than saturated or trans-fats; the addition of trans fatty acid sources should be clearly labelled. Fortunately, the EU Commission, 2019, has now adopted a regulation setting a maximum limit for trans-fats in industrially produced trans-fat of 2 g/100 g of fat [45]. Some isolated starches could be replaced by plants naturally low in phenylalanine such as cassava. In SLPFs, added sugar should be restricted if protein-containing comparators do not contain it. It is hypothesised that high sugar consumption may affect gut microbiota, disturbing the crosstalk between the gut and systemic metabolism, with a potentially harmful impact on metabolic health [46]. Reducing the salt content of some savoury products and replacing it with herbs and spices to improve or maintain the taste and flavour of SLPFs would be beneficial. A simple traffic light colour system has been proposed to categorise SLPFs based on their nutritional profile [10] and this may help patients reduce refined carbohydrate and salt intake and increase their consumption of healthier fats and complex carbohydrates.

In this evaluation of SLPFs, difficulties in accessing nutritional composition data has led to several limitations. Data was missing for some key nutrients such as fibre. Nutritional values were often reported as “<0.5” or “<0.1”, and so the precise content was unclear. There were occasional discrepancies in nutritional information between SLPFs and regular foods. Some foods provided information for dried ingredients whilst others only for cooked/prepared products. The selection of protein-containing foods as comparators and how the products were grouped was subjective. Finally, this study only examined products accessible on UK prescription compared with protein-containing products available from UK supermarkets. Detailed nutritional composition analysis of SLPFs available on prescription compared with regular equivalent products in other countries is warranted to determine if findings are consistent.

## 5. Conclusions

In conclusion, this UK study shows that the nutritional content of SLPFs available on ACBS prescription differed to regular comparable foods but with no clear consistent pattern. Almost two thirds of SLPF subgroups contained less total fat but with palm oil and hydrogenated vegetable oil as key fat sources. Over two thirds of SLPF subgroups contained more carbohydrate commonly as isolated starches. More added fibre was identified in SLPFs but predominantly in the form of hydrocolloids. It is possible that habitual consumption of SLPFs higher in salt, sugars, isolated starches, or saturated fat may contribute to future nutritional comorbidities.

Stricter nutritional composition regulations, improvements in product labelling and access to full nutritional composition data will allow health professionals and patients to make informed decisions when prescribing and using SLPFs. Identifying upper limits for macronutrients, and improving fat and carbohydrate sources is essential in supporting patients with PKU in meeting their nutritional needs and improving health outcomes.

## Figures and Tables

**Figure 1 nutrients-12-01893-f001:**
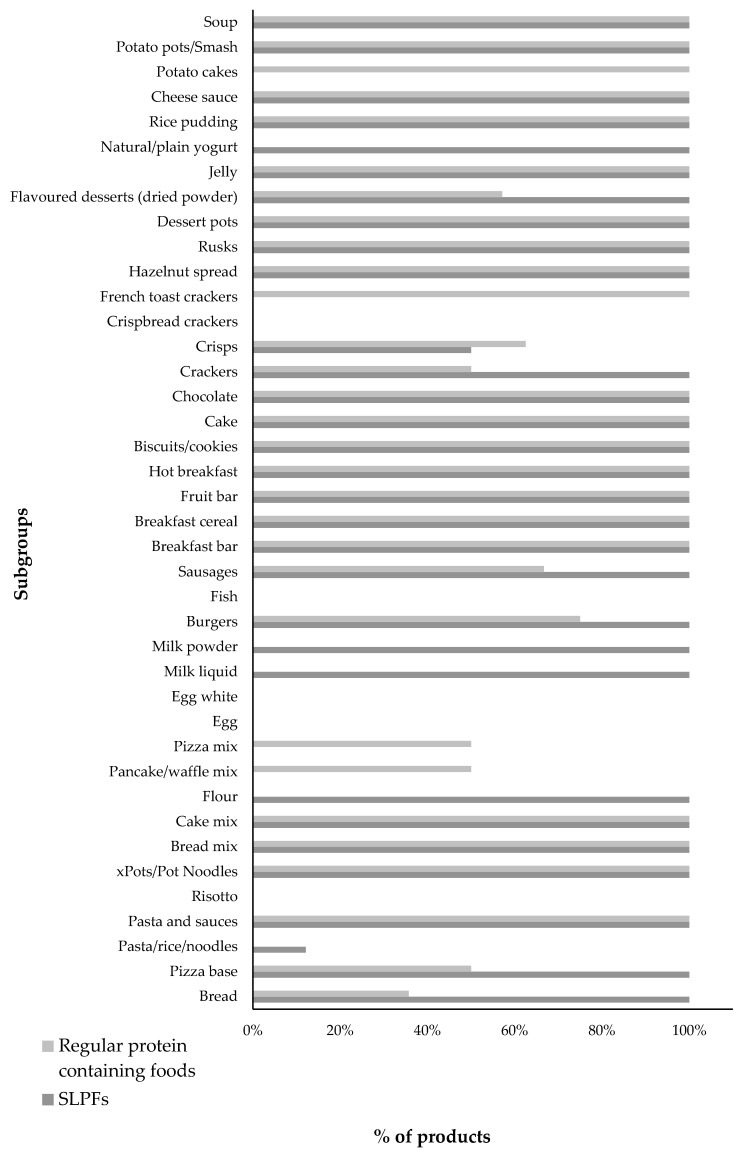
Percentage of regular and special low protein food (SLPF) products containing added sugar in their ingredient list by subgroup.

**Figure 2 nutrients-12-01893-f002:**
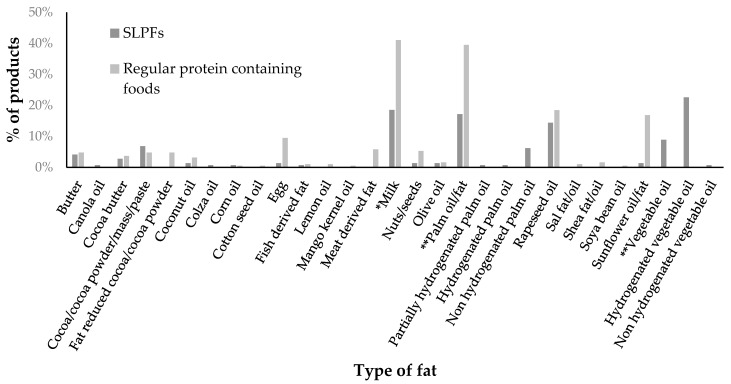
Percentage of SLPFs and regular protein containing foods containing different types of fat in their ingredient lists. * Not including milk protein (where products specified this as an ingredient) ** oil/fat, did not specify whether it was hydrogenated or non-hydrogenated.

**Figure 3 nutrients-12-01893-f003:**
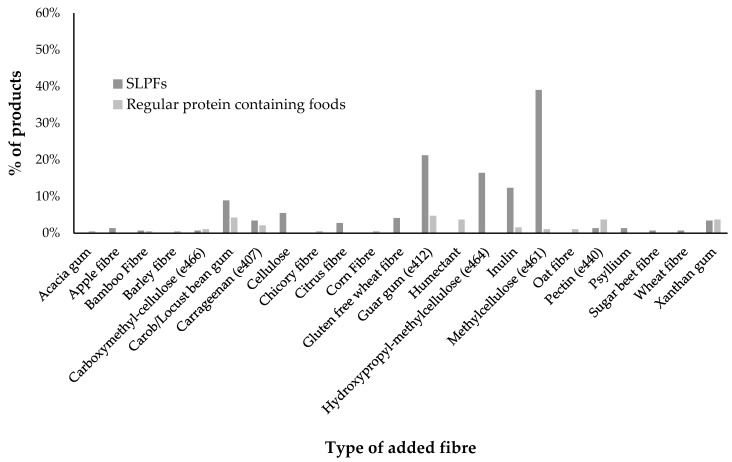
Percentage of regular and SLPF products containing added fibre in their ingredient lists by type of fibre.

**Table 1 nutrients-12-01893-t001:** Phenylalanine content and identified sources of natural protein for all special low protein food (SLPF) subgroups. Values displayed as mean (range).

SLPF Subgroup	Phenylalanine (mg) per 100 g of Product	Identified Sources of Natural Protein/Phenylalanine in Each SLPF Subgroup
Bread (*n* = 13)	15 (8–30)	Yeast (*n* = 13), fennel seeds (*n* = 1), anis seeds (*n* = 1)
Pizza base (*n* = 2)	13 (2–24)	Yeast (*n* = 2)
Pasta/rice/noodles (*n* = 33)	13 (8–25)	Rice flour (*n* = 5)
Pasta and sauces (prepared) (*n* = 5)	8 (3–14)	Milk (*n* = 4), yeast extract (*n* = 1), cheese powder (*n* = 1)
Risotto (*n* = 1)	6	Milk (*n* = 1)
xPots/pot noodles (prepared) (*n* = 4)	9 (6–15)	Peas (dried) (*n* = 1), milk (*n* = 4)
Bread mix (*n* = 3)	15 (4–20)	Yeast (*n* = 1)
Cake mix (*n* = 4)	14 (4–30)	Cocoa powder (*n* = 1), cocoa (*n* = 1)
Flour (*n* = 4)	5 (4–<10)	No sources identified
Pancake/waffle mix (*n* = 1)	22	No sources identified
Pizza mix (*n* = 1)	<31	No sources identified
Egg replacer (dried mix) (*n* = 3)	7 (<5–10)	No sources identified
Egg white replacer (*n* = 1)	Nil added	No sources identified
Milk (liquid) (*n* = 4)	6 (0–10)	Milk (*n* = 4), whey powder (*n* = 2)
Milk (powder) (*n* = 1)	20	Milk (*n* = 1), whey permeate (*n* = 1)
Burgers (prepared) (*n* = 3)	25 (16–31)	Milk (*n* = 2), yeast (*n* = 1)
Fish substitute (prepared) (*n* = 1)	38	Shrimps (*n* = 1), cod (*n* = 1), rice flour (*n* = 1), milk (*n* = 1)
Sausages (prepared) (*n* = 3)	33 (29–38)	Milk (*n* = 3), potato flake (*n* = 3)
Breakfast bar (*n* = 4)	17 (12–25)	Milk (*n* = 4), cocoa powder (*n* = 1)
Breakfast cereal (dried) (*n* = 3)	12 (6–22)	Cocoa powder (*n* = 1)
Fruit bar (*n* = 1)	16	Egg (*n* = 1)
Hot breakfast cereal (prepared with water) (*n* = 4)	4 (2–6)	Cocoa powder (*n* = 1), milk (*n* = 4)
Biscuits/cookies (*n* = 9)	10 (1–27)	Cocoa mass (*n* = 1), egg (*n* = 1), cocoa (*n* = 2)
Cake (*n* = 3)	6 (6–6)	No sources identified
Chocolate (*n* = 2)	12 (<10–14)	Milk (*n* = 1), cocoa powder (*n* = 1), carob flour (*n* = 1)
Crackers (*n* = 3)	12 (10–17)	No sources identified
Crisps (*n* = 4)	16 (8–22)	Wheat flour (*n* = 2), rice flour (*n* = 1), whey powder (*n* = 2), yeast extract powder (*n* = 1), cheese powder (*n* = 1), yeast powder (*n* = 1)
Crispbread crackers (*n* = 1)	6	Pea starch (*n* = 1)
French toast crackers (*n* = 1)	30	Baker’s yeast (*n* = 1)
Hazelnut spread (*n* = 1)	19	Milk (*n* = 1), hazelnuts (*n* = 1), almonds (*n* = 1), cocoa paste (*n* = 1)
Rusks (*n* = 1)	4	Milk (*n* = 1)
Dessert pot (*n* = 2)	<4	No sources identified
Flavoured desserts (prepared) (*n* = 4)	5 (1–13)	Milk (*n* = 4), chocolate powder (*n* = 1)
Jelly (dried) (*n* = 2)	<2	No sources identified
Rice pudding (*n* = 4)	6 (5–8)	Milk (*n* = 4)
Yogurt (prepared) (*n* = 1)	2	No sources identified
Cheese sauce (prepared) (*n* = 1)	13	Milk (*n* = 1)
Potato cakes (prepared) (*n* = 1)	46	Potato flake (*n* = 1)
Potato pots/Smash (prepared) (*n* = 3)	25 (23–27)	Potato flake (*n* = 3), milk (*n* = 3)
Soup (prepared) (*n* = 4)	2 (1–2)	Milk (*n* = 4), peas (*n* = 2)

**Table 2 nutrients-12-01893-t002:** Key sources of added sugar identified from ingredient lists for SLPFs and regular protein-containing foods.

Key Sources of Added Sugar	% of SLPF (*n* = 146)	% of Regular Protein Containing Foods (*n* = 190)
Sugar	52% (*n* = 76/146)	58% (*n* = 111/190)
Glucose	29% (*n* = 43/146)	23% (*n* = 44/190)
Maltodextrin	23% (*n* = 33/146)	13% (*n* = 25/190)
Dextrose	15% (*n* = 22/146)	12% (*n* = 22/190)
Sucrose	3% (*n* = 5/146)	1% (*n* = 2/190)
Fructose	<1% (*n* = 1/146)	6% (*n* = 12/190)

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
