# Peer review of "Special Low Protein Foods in the UK: An Examination of Their Macronutrient Composition in Comparison to Regular Foods"

_nutrients, 2020, doi:10.3390/nu12061893_

Round 1

Reviewer 1 Report

The present study provides a nutritional purview of special low protein foods in the UK market as that are used by individuals with PKU.  They were compared to conventional foods in the marketplace.  These types of analyses are important as foods for special uses are critical for the management of diseases such as PKU or celiac disease, and should be palatable as well as fundamentally healthy to prevent risk for other diet-related ailments.

I do have a few suggestions for the authors moving forward.

1.  Within the methods, can more clarity for how "regular" protein foods were selected as comparators to the SLPFs.  Was this systematic? or random?  where there any controls in place to prevent foods with a very different nutritional composition chosen, which may not represent the majority of regular foods?

2.  Line 223.  "with PKU has been noted"  Be cognizant of the tense of the manuscript, and use past tense as often as possible

3. The latter sections of the Discussion begin read as a commentary, outlining what "should be done"  I suggest consolidating these 2 paragraphs (Lines 252-277), as an the opportunity for industry and regulations to require some nutritional guidelines for these types of products.  Analysis on dietary patterns where these foods are consumed relative to nutritional guidelines would likely be required to inform what "should" be implemented as policy for SLPFs.

4.  I would caution the authors in making absolute statements around the sugar and microbiome.  This is an emerging area of research, and disagree that the evidence unanimously demonstrates that sugar "has a profound impact on microbiota...."  I suggest the language here be more suggestive and align with the state of the science.

5. The conclusion reads as if it was written for a commentary.  The conclusion should provide a broad overview of what the major findings of the study were .  The conclusion currently touches broad themes which were only discussed in the Discussion section. Some of this can remain intact, where the authors suggest are areas around future research and policy evaluation could be considered.  But the conclusion of the article should primarily touch upon what the study conclusions are.  I suggest the Conclusion be re-worked.

Author Response

Response to peer reviewer 1

Thank you so much for taking the time to review our manuscript. We really appreciate your feedback and comments. Please see our responses below (line numbers correspond to the lines on the ‘tracked changes’ word document/manuscript):

  1. Within the methods, can more clarity for how "regular" protein foods were selected as comparators to the SLPFs.  Was this systematic? or random?  where there any controls in place to prevent foods with a very different nutritional composition chosen, which may not represent the majority of regular foods?

Response: Thank you. We have added in extra detail to the methodology. See line 83-85:

‘Taste, texture, recipe ingredients and purpose were considered when choosing comparator foods. Where possible, only regular products that had nutritional analysis available in the same format to SLPFs were considered e.g. dried format or after preparation.’

  1. "with PKU has been noted"  Be cognizant of the tense of the manuscript, and use past tense as often as possible

Response: Now on line 258 we have changed from ‘…with PKU is noted’ to ‘…with PKU has also been described.

  1. The latter sections of the Discussion begin read as a commentary, outlining what "should be done"  I suggest consolidating these 2 paragraphs (Lines 252-277), as an the opportunity for industry and regulations to require some nutritional guidelines for these types of products.  Analysis on dietary patterns where these foods are consumed relative to nutritional guidelines would likely be required to inform what "should" be implemented as policy for SLPFs.

Response: Thank you. We agree that further analysis on dietary patterns is warranted and would help inform what should be implemented as policy and have added this to the discussion. However, with SLPFs even without data on dietary patterns, we consider that some of the nutritional composition and labelling of the SPLFs can still be significantly improved by the manufacturers. We have made changes to the manuscript which have incorporated the reviewer’s comments. See below and lines 302-328.

‘Overall, there is limited research into the dietary patterns of patients with PKU, but evidence suggests SLPFs contribute up to 47% of energy intake [11]. Many contemporary low phenylalanine protein substitutes have a low fat and carbohydrate content, meaning there is an increased reliance on SLPFs to provide these macronutrients [42,43]. With a ‘treatment for life’ policy, it is essential that SLPFs have a nutritional profile that supports long term healthy eating patterns.

There are many recommendations required to improve standards in the nutritional composition and labelling of UK SLPFs. Transparency is necessary by SLPF manufacturers about the nutritional profile of their products. All ingredients should be clearly listed including sources of, at least, starch, sugar, fat and fibre and the amount of fibre added (per 100 g/100 ml) for all SLPFs. Nutritional analysis for both dried and prepared weights should be available. Packaging and website nutritional information should be accurate and consistent. To ensure that all SLPFs can be safely consumed without calculation and measurement, the phenylalanine content should be no more than 25 mg/100 g for all prescribed SLPFs; and no more phenylalanine than 5 mg/100 ml for milk replacements [44]. SLPF macronutrient composition regulations should be strengthened, ensuring similarity to regular protein containing comparators. Upper limits should be set for carbohydrate and fat content. Fat sources should be predominantly poly- or mono-unsaturated rather than saturated or trans-fats; the addition of trans fatty acids sources should be clearly labelled. Fortunately, the EU Commission, 2019, have now adopted a regulation setting a maximum limit for trans-fats in industrially produced trans-fat of 2g/100 g of fat [45]. Some isolated starches could be replaced by plants naturally low in phenylalanine such as cassava. In SLPFs, added sugar should be restricted if protein containing comparators do not contain it. It is hypothesised that high sugar consumption may affect gut microbiota, disturbing the crosstalk between the gut and systemic metabolism, with potential harmful impact on metabolic health [46]. Reducing the salt content of some savoury products and replacing with herbs and spices to improve or maintain the taste/flavour of certain SLPFs would be beneficial. A simple traffic light colour system has been proposed to categorise SLPFs based on their nutritional profile [10] and this may help patients reduce refined carbohydrate and salt intake and increase their consumption of healthier fats and complex carbohydrates.’

  1. I would caution the authors in making absolute statements around the sugar and microbiome.  This is an emerging area of research and disagree that the evidence unanimously demonstrates that sugar "has a profound impact on microbiota...."  I suggest the language here be more suggestive and align with the state of the science.

Response: Thank you. We have altered this sentence to ‘It is hypothesised that high sugar consumption may affect gut microbiota, disturbing’ (line 322-323).

  1. The conclusion reads as if it was written for a commentary.  The conclusion should provide a broad overview of what the major findings of the study were .  The conclusion currently touches broad themes which were only discussed in the Discussion section. Some of this can remain intact, where the authors suggest are areas around future research and policy evaluation could be considered.  But the conclusion of the article should primarily touch upon what the study conclusions are.  I suggest the Conclusion be re-worked.

Response: Thank you. We have re-written the conclusion following the reviewer’s comments. See below and lines 340-358.

In conclusion, this UK study shows that the nutritional content of SLPFs available on ACBS prescription differed to regular comparable foods but with no clear consistent pattern. Almost two thirds of SLPF subgroups contained less total fat but with palm oil and hydrogenated vegetable oil key fat sources. Over two thirds of SLPF subgroups contained more carbohydrate commonly as isolated starches. More added fibre was identified in SLPFs but predominantly in the form of hydrocolloids. It is possible that habitual consumption of SLPFs higher in salt, sugars, isolated starches, or saturated fat may contribute to future nutritional comorbidities.

Stricter nutritional composition regulations, improvements in product labelling and access to full nutritional composition data, will allow health professionals and patients to make informed decisions when prescribing and using SLPFs. Identifying upper limits for macronutrients, and improving fat and carbohydrate sources is essential in supporting patients with PKU in meeting their nutritional needs and improving health outcomes.’

Reviewer 2 Report

Paper is well written and obviously represents a sizeable amount of work. 

I do have 2 suggestions. It was noted that the phenylalanine content of "regular" foods was not compared to the low protein foods. There are now good sources for this information, and any foods not having this data available can be extrapolated from protein content (and noted as such). Since this is the reason for development of low protein foods, I think this information would be important to include.

My second suggestion is tangental but maybe important to at least acknowledge. A PKU diet restricted in intact protein depends for nutritional completeness on medical food/supplement nutrients (75% on average). This complicates drawing any health risk associations with low protein food consumption alone since nutrients from those sources are not as predominant a source of nutrients as the medical food. I realize this is beyond the scope of this paper, but believe the percentage of overall nutrients contributed by low protein foods should be acknowledged, as well as the understanding that many PKU patients have poor compliance with dietary restrictions, and medical food/supplement consumption. This alone makes drawing health risk conclusions complicated.

Author Response

Response to peer reviewer 2

Thank you so much for taking the time to review our manuscript. We really appreciate your feedback and comments. Please see our responses below (line numbers correspond to the lines on the ‘tracked changes’ word document/manuscript):

  1. I do have 2 suggestions. It was noted that the phenylalanine content of "regular" foods was not compared to the low protein foods. There are now good sources for this information, and any foods not having this data available can be extrapolated from protein content (and noted as such). Since this is the reason for development of low protein foods, I think this information would be important to include.

    Response:
    Thank you.We have added this information into the table in the appendix and incorporated it into the results section, merging the protein and phenylalanine subsections together. See appendix and line 109-113 and below:

‘All SLPFs contained between 43-100% less protein and 60-100% less phenylalanine than regular foods. Table 1 displays the mean and range for phenylalanine content and sources of phenylalanine for all SLPF subgroups. The main sources of phenylalanine found in SLPFs were milk (including milk protein) (32% of SLPFs; n=47/146) and yeast (14% of SLPFs; n=21/146). For 91% of SLPFs (n=133/146), the phenylalanine content was ≤25 mg per 100 g (table 1).’

 We have also added an explanation into the methods about how this information was estimated from protein content. See line 82-83 and below:

‘Phenylalanine content was estimated by calculating that 1 g of protein contained 50 mg phenylalanine [12].’

  1. My second suggestion is tangential but maybe important to at least acknowledge. A PKU diet restricted in intact protein depends for nutritional completeness on medical food/supplement nutrients (75% on average). This complicates drawing any health risk associations with low protein food consumption alone since nutrients from those sources are not as predominant a source of nutrients as the medical food. I realize this is beyond the scope of this paper, but believe the percentage of overall nutrients contributed by low protein foods should be acknowledged, as well as the understanding that many PKU patients have poor compliance with dietary restrictions, and medical food/supplement consumption. This alone makes drawing health risk conclusions complicated.

    Response: Thank you for this interesting comment. We agree with this thought, however with modern protein substitutes now containing little carbohydrate and fat for patients over 3 years of age, there is more reliance on SLPFs to provide these macronutrients. We are also aware that protein substitutes are usually the major source of vitamins and minerals for PKU patients, but micronutrients were not the focus of this paper. With regards to the comment on drawing health risk conclusions, we have tried to be more suggestive as opposed to making direct conclusions and have altered some of the wording within the discussion around this. See below and the manuscript from line 302 onwards.

‘Overall, there is limited research into the dietary patterns of patients with PKU, but evidence suggests SLPFs contribute up to 47% of energy intake [11]. Many contemporary low phenylalanine protein substitutes have a low fat and carbohydrate content, meaning there is an increased reliance on SLPFs to provide these macronutrients [42,43]. With a ‘treatment for life’ policy, it is essential that SLPFs have a nutritional profile that supports long term healthy eating patterns.

There are many recommendations required to improve standards in the nutritional composition and labelling of UK SLPFs. Transparency is necessary by SLPF manufacturers about the nutritional profile of their products. All ingredients should be clearly listed including sources of, at least, starch, sugar, fat and fibre and the amount of fibre added (per 100 g/100 ml) for all SLPFs. Nutritional analysis for both dried and prepared weights should be available. Packaging and website nutritional information should be accurate and consistent. To ensure that all SLPFs can be safely consumed without calculation and measurement, the phenylalanine content should be no more than 25 mg/100 g for all prescribed SLPFs; and no more phenylalanine than 5 mg/100 ml for milk replacements [44]. SLPF macronutrient composition regulations should be strengthened, ensuring similarity to regular protein containing comparators. Upper limits should be set for carbohydrate and fat content. Fat sources should be predominantly poly- or mono-unsaturated rather than saturated or trans-fats; the addition of trans fatty acids sources should be clearly labelled. Fortunately, the EU Commission, 2019, have now adopted a regulation setting a maximum limit for trans-fats in industrially produced trans-fat of 2g/100 g of fat [45]. Some isolated starches could be replaced by plants naturally low in phenylalanine such as cassava. In SLPFs, added sugar should be restricted if protein containing comparators do not contain it. It is hypothesised that high sugar consumption may affect gut microbiota, disturbing the crosstalk between the gut and systemic metabolism, with potential harmful impact on metabolic health [46]. Reducing the salt content of some savoury products and replacing with herbs and spices to improve or maintain the taste/flavour of certain SLPFs would be beneficial. A simple traffic light colour system has been proposed to categorise SLPFs based on their nutritional profile [10] and this may help patients reduce refined carbohydrate and salt intake and increase their consumption of healthier fats and complex carbohydrates.

In this evaluation of SLPFs, difficulties in accessing nutritional composition data led to several limitations. Data was missing for some key nutrients such as fibre. Nutritional values were often reported as ‘<0.5’ or ‘<0.1’, and so the precise content was unclear. There were occasional discrepancies in nutritional information between SLPFs and regular foods. Some foods provided information for dried ingredients whilst others only for cooked/prepared products. The selection of protein containing foods as comparators and how the products were grouped was subjective. Finally, this study only examined products accessible on UK prescription compared with protein containing products available from UK supermarkets. Detailed nutritional composition analysis of SLPFs available on prescription compared with regular equivalent products in other countries is warranted to determine if findings are consistent.

5. Conclusions
In conclusion, this UK study shows that the nutritional content of SLPFs available on ACBS prescription differed to regular comparable foods but with no clear consistent pattern. Almost two thirds of SLPF subgroups contained less total fat but with palm oil and hydrogenated vegetable oil key fat sources. Over two thirds of SLPF subgroups contained more carbohydrate commonly as isolated starches.
More added fibre was identified in SLPFs but predominantly in the form of hydrocolloids. It is possible that habitual consumption of SLPFs higher in salt, sugars, isolated starches, or saturated fat may contribute to future nutritional comorbidities.

Stricter nutritional composition regulations, improvements in product labelling and access to full nutritional composition data, will allow health professionals and patients to make informed decisions when prescribing and using SLPFs. Identifying upper limits for macronutrients, and improving fat and carbohydrate sources is essential in supporting patients with PKU in meeting their nutritional needs and improving health outcomes.’